# Implementation of CT Coronary Angiography as an Alternative to Invasive Coronary Angiography in the Diagnostic Work-Up of Non-Coronary Cardiac Surgery, Cardiomyopathy, Heart Failure and Ventricular Arrhythmias

**DOI:** 10.3390/jcm10112374

**Published:** 2021-05-28

**Authors:** Thomas P. W. van den Boogert, Bimmer E. P. M. Claessen, Adrienne van Randen, Joost van Schuppen, S. Matthijs Boekholdt, Marcel A. M. Beijk, M. Karlijn Vrijmoeth, Jan Baan, M. Marije Vis, Jacobus A. Winkelman, Antoine H. G. Driessen, Jaap Stoker, R. Nils Planken, Jose P. Henriques

**Affiliations:** 1Part of the Amsterdam Cardiovascular Sciences, Heart Centre, Amsterdam UMC, University of Amsterdam, 1105AZ Amsterdam, The Netherlands; t.p.vandenboogert@amsterdamumc.nl (T.P.W.v.d.B.); s.m.boekholdt@amsterdamumc.nl (S.M.B.); m.a.beijk@amsterdamumc.nl (M.A.M.B.); m.k.vrijmoeth@amsterdamumc.nl (M.K.V.); j.baan@amsterdamumc.nl (J.B.); m.m.vis@amsterdamumc.nl (M.M.V.); j.a.winkelman@amsterdamumc.nl (J.A.W.); a.h.driessen@amsterdamumc.nl (A.H.G.D.); 2Department of Cardiology, Noordwest Ziekenhuisgroep, 1815JD Alkmaar, The Netherlands; bimmerclaessen@gmail.com; 3Amsterdam Cardiovascular Sciences, Department of Radiology and Nuclear Medicine, Amsterdam UMC, University of Amsterdam, 1105AZ Amsterdam, The Netherlands; a.vanranden@amsterdamumc.nl (A.v.R.); j.vanschuppen@amsterdamumc.nl (J.v.S.); r.n.planken@amsterdamumc.nl (R.N.P.); 4Amsterdam Gastroenterology Endocrinology Metabolism, Department of Radiology and Nuclear Medicine, Amsterdam UMC, University of Amsterdam, 1105AZ Amsterdam, The Netherlands; j.stoker@amsterdamumc.nl

**Keywords:** computed tomography angiography, coronary artery disease, percutaneous coronary intervention, coronary angiography

## Abstract

To assess the need for additional invasive coronary angiography (CAG) after initial computed tomography coronary angiography (CTCA) in patients awaiting non-coronary cardiac surgery and in patients with cardiomyopathy, heart failure or ventricular arrhythmias, and to determine differences between patients that were referred to initial CTCA or direct CAG, consecutive patients were included between August 2017 and January 2020 and categorized as those referred to initial CTCA (conform protocol), and to direct CAG (non-conform protocol). Out of a total of 415 patients, 78.8% (327 patients, mean age: 57.9 years, 67.3% male) were referred to initial CTCA, of whom 260 patients (79.5%) had no obstructive lesions (<50% DS). A total of 55 patients (16.8%) underwent additional CAG after initial CTCA, which showed coronary lesions of >50% DS in 21 patients (6.3% of 327). Eighty-eight patients (mean age: 66.0 years, 59.1% male) were directly referred to CAG (non-conform protocol). These patients were older and had more cardiovascular risk factors compared to patients that underwent initial CTCA (conform protocol), and coronary lesions of >50% DS were detected in 16 patients (17.2%). Revascularization procedures were infrequently performed in both groups: initial CTCA (3.0%), direct CAG (3.4%). The use of CTCA as a gatekeeper CAG in the diagnostic work-up of non-coronary cardiac surgery, cardiomyopathy, heart failure and ventricular arrhythmias is feasible, and only 17% of these patients required additional CAG after initial CTCA. Therefore, CTCA should be considered as the initial imaging modality to rule out CAD in these patients.

## 1. Introduction

Coronary angiography (CAG) is the reference standard to diagnose obstructive coronary artery disease (CAD). However, this invasive diagnostic procedure is associated with discomfort, pain and the risk of adverse events such as myocardial infarction, stroke, arterial thrombosis and dissection or bleeding [1]. A non-invasive method to diagnose CAD is computed tomography coronary angiography (CTCA), which offers a high negative predictive value of up to 99.0% [2]. Besides the use of CTCA in patients with chest pain and suspected CAD, the most recent European Society of Cardiology (ESC) guidelines recommend CTCA as an alternative for CAG to rule out CAD in patients awaiting non-coronary cardiac surgery and in patients with cardiomyopathy, heart failure or ventricular arrhythmias, but only in patients with a low risk of CAD [3,4]. However, currently, these patients almost exclusively undergo CAG to rule out CAD, despite having a generally low diagnostic yield, as only ±20% of patients awaiting non-coronary cardiac surgery have obstructive coronary lesions [5].

It is, however, still a challenge to implement CTCA for these indications in daily clinical care. Therefore, we studied the implementation of CTCA as the initial imaging modality to exclude CAD in the diagnostic work-up for non-coronary cardiac surgery, cardiomyopathy, heart failure and ventricular arrhythmias. Our hypothesis was that implementation of CTCA as a gatekeeper for CAG was feasible and that only a minority of patients would require additional CAG. To test this hypothesis, we registered clinical care since CTCA implementation in August 2017 and assessed the number of CTCA and CAG procedures over the course of 30 months. In this study, we aimed to assess the need for additional CAG after the initial CTCA and to determine differences between patients that were referred to initial CTCA (conform protocol) or initial CAG (non-conform protocol).

## 2. Materials and Methods

### 2.1. Study Design and Population

For this observational cohort study, we included patients from 1 August 2017 until 31 January 2020 with a follow-up duration of 6 months. At the start of the project, CTCA was implemented as the standard of care to exclude CAD in patients without known obstructive CAD or typical anginal complaints in the diagnostic work-up for non-coronary cardiac surgery, cardiomyopathy, heart failure and ventricular arrhythmias (excluding ventricular fibrillation after cardiac arrest). This study included patients that would typically undergo invasive CAG before the start of the project. If CTCA indicated obstructive CAD, a subsequent CAG was performed. Atrial fibrillation was not an exclusion criterion. However, the CTCA acquisition protocol was adjusted accordingly. Patients who could not undergo a CT scan because of renal failure (eGFR < 30 mL/min/1.73 m^2^) or contrast allergy were excluded. Patient management was at the discretion of the treating clinician. Revascularization was at the discretion of the interventional cardiologist, based on invasive CAG with fractional flow reserve (FFR). In order to assist this potentially challenging implementation process, a dedicated physician was positioned to coordinate this process (TvdB). Cardiologists were actively encouraged to refer patients for initial CTCA, and all CAG orders were evaluated. If patients were suitable for CTCA, the referring cardiologist was consulted to evaluate the patient’s eligibility to convert the CAG order to CTCA. The local institutional review board approved this study, and all patients signed informed consent for this study. This project was funded by an institutional innovation grant, reference number #2017-09.

### 2.2. CTCA Acquisition Protocol

All CTCA scans were performed using a third-generation dual source CT scanner (Somatom Force, Siemens Healthcare, Erlangen, Germany). Sublingual nitroglycerine spray was administered before CTCA examination, and oral beta blockers were administered if the heart rate was >65 per min. The scan delay was determined using a test bolus of 10 mL undiluted contrast medium (Ultravist 300: iopromide 300 mg I/mL, Bayer AG, Leverkusen, Germany) and a fixed kV setting of 100 kV, after which four seconds were added for the scan delay of the main bolus. CT scanner acquisition parameters were: detector collimation 2 × 96 × 0.6 mm, slice acquisition 2 × 192 × 0.6 mm by means of a z-flying focal spot, gantry rotation time of 250 ms, temporal resolution of 66 ms, 70–120 kV tube voltage (CARE kV) and 180–600 μA tube current. High-pitch spiral scanning was performed in diastole in patients with a regular heart rate of <70/min. For patients with irregular heart rates or heart rates of > 70/min, a prospective sequential scan was performed in diastole and for heart rates of > 80/min in systole. For contrast delivery, we used a patient-tailored contrast delivery protocol adjusting the iodine delivery rate (IDR) via saline dilution based on body weight and kV settings (IDR scheme can be found in Appendix A). Images were reconstructed with a slice thickness of 0.6 mm and an increment of 0.4 mm using iterative reconstruction factor 2 (ADMIRE, Siemens Healthcare, Erlangen, Germany). All coronary segments with a diameter of >1.5 mm were visually evaluated for the presence of coronary stenosis and graded according to the standardized Coronary Artery Disease—Reporting and Data System (CAD-RADS) method [6].

### 2.3. CAG Acquisition Protocol

Angiograms were performed in accordance with local practice, and CAD was assessed by the performing cardiologist by visual estimation. Following sublingual or intracoronary administration of nitroglycerine, angiography of the left and right coronary arteries was performed for at least two orthogonal views showing all segments free of foreshortening or vessel overlap. Contrast (Xenetix 300 Iobitridol 658 mg/mL) administration was performed by manual injection.

### 2.4. Data Collection and Definitions

The collected data consisted of patient demographics, patient comorbidities and procedural characteristics. Demographic characteristics included information on gender, age and body mass index (BMI). Patient comorbidities included diabetes, dyslipidemia, atrial fibrillation, hypertension and family history of CAD. We defined impaired renal function according to eGFR categories G3a/G3b (eGFR 30–60 mL/min/1.73 m^2^). Patients with eGFR < 30 mL/min/1.73 m^2^ were excluded. Procedural characteristics included the degree of diameter stenosis (DS) per vessel for either CTCA or CAG and categorized as: 0–50%, 50–70%, >70% and total occlusion. Post-procedural data included revascularization with either percutaneous coronary intervention (PCI) or coronary artery bypass grafting (CABG).

### 2.5. Statistical Analysis

All statistical analyses were performed using R software version 3.5.1 (R Foundation for Statistical Computing, Vienna, Austria). Continuous baseline variables are presented as means ± standard deviations. Categorical baseline variables are displayed as frequencies and percentages. Differences between groups were compared using the chi-square test for frequencies and Student’s t-test for continuous variables.

## 3. Results

### 3.1. Clinical Characteristics and CTCA Findings of Patients That Underwent Initial CTCA

Out of a total of 415 patients, 327 patients (78.8%) underwent initial CTCA, with a mean age of 57.9 ± 14.0 years, where 67.3% were male (Table 1). Cardiovascular comorbidities included atrial fibrillation (26.6%), impaired renal function (eGFR 30–60 mL/min/1.73 m^2^) (13.5%), diabetes mellitus (12.3%), dyslipidemia (34.0%), hypertension (39.9%), current smokers (20.4%) and family history of CAD (42.4%). The majority of CTCA scans were performed in patients awaiting non-coronary cardiac surgery (*n* = 163, 49.8%). The other CTCA scans were performed in the diagnostic work-up for ventricular arrhythmias (*n* = 76, 23.2%), various types of cardiomyopathy (*n* = 57, 17.4%) and heart failure (*n* = 31, 9.5%).

Different indications for CTCA were associated with differences in patient characteristics. For example, age ranged between 55.1 years old in patients undergoing diagnostic work-up for ventricular arrhythmias and 61.6 years old in patients undergoing diagnostic work-up for heart failure (Table 1). As it was to be expected with lower age, atrial fibrillation and impaired renal function (eGFR 30–60 mL/min/1.73 m^2^) were less common in patients undergoing work-up for ventricular arrhythmias, as compared to the patients awaiting non-coronary cardiac surgery (*p* = 0.01 and *p* = 0.004, respectively).

Of the 327 patients that underwent initial CTCA, 260 patients (79.5%) had no or non-obstructive CAD (<50% DS), 30 patients (9.2%) had coronary lesions of 50–70% and 23 patients (7.0%) had coronary lesions of >70% (Table 2). A total of 14 patients (4.3%) had a non-diagnostic CTCA scan (Table 2). Reasons for non-diagnostic scans included motion artefacts (*n* = 11), blooming artefacts due to extensive calcification that prevented diagnosis (*n* = 2) and insufficient coronary attenuation (*n* = 1, coronary attenuation was 198 HU, kV settings were 120 kV). Eight patients with motion artefacts were known with atrial fibrillation; the other 69 patients that were known with atrial fibrillation had diagnostic CTCA examinations. Of the 14 patients with a non-diagnostic CTCA scan, 6 underwent additional CAG. Of the remaining eight patients, three patients were accepted for non-coronary cardiac surgery without additional CAG, one patient underwent additional non-invasive functional testing without signs of ischemia and in four patients, the cardiologist elected a watchful waiting approach.

### 3.2. The Need for Additional CAG after Initial CTCA and Subsequent Revascularization

Of the 327 patients who underwent initial CTCA, 55 patients underwent additional CAG after CTCA: six because the initial CTCA was non-diagnostic, and 49 because the initial CTCA suggested obstructive CAD. In these 55 patients, additional CAG showed lesions of <50% DS in 34 patients (61.8%) and confirmed obstructive lesions in 21 patients (38.2%). From these 21 patients, 10 patients (18.2%) had lesions of 50–70% DS, and 11 patients (20.0%) had lesions of >70% DS. Ten patients underwent revascularization procedures (CABG *n* = 3, PCI *n* = 7), of whom there were five patients awaiting non-coronary cardiac surgery (three CABG, two PCI), two patients with cardiomyopathy (PCI) and three with ventricular arrhythmia (PCI). In these ten patients, the preceding CTCA showed lesions of 50–70% DS in two patients and lesions of >70% DS in eight patients.

### 3.3. Comparison of Stenosis Grading with Initial CTCA and Additional CAG

A total of 49 CTCA and CAG datasets were available for stenosis grading comparison instead of 55 due to non-diagnostic CTCA scans in six patients. In these 49 patients, CAG and CTCA revealed the same degree of stenosis in 30 (61.2%) patients, and CTCA overestimated the degree of stenosis in 19 patients (38.8%). In ten patients, CTCA indicated a 50–70% DS lesion, while CAG indicated a <50% DS lesion. In nine patients, CTCA indicated a >70% DS lesion, while CAG indicated a 50–70% DS lesion (*n* = 3) or <50% DS lesion (*n* = 6). In no patients did CTCA underestimate the degree of stenosis compared with angiographic assessment.

### 3.4. Clinical Characteristics and Imaging Findings of Patients Who Underwent Direct CAG vs. Initial CTCA

A total of 88 patients were directly referred for CAG and were considered for the non-conform protocol (Table 3). Patients who were directly referred for CAG were generally older (66.0 vs. 57.9 years old, *p* < 0.001), had a higher BMI (27.5 vs. 26.2 kg/m^2^, *p* = 0.03) and were significantly more likely to have impaired renal function (eGFR 30–60 mL/min/1.73 m^2^) (26.4% vs. 13.5%, *p* = 0.01) (Table 3). Invasive CAG showed non-obstructive CAD (<50% DS) in 72 patients (81.8%), lesions with 50–70% DS in 16 patients (18.2%) and lesions with DS > 70% in 6 patients (6.8%) (Table 2). Of the 88 patients who underwent direct CAG, a total of 3 patients underwent revascularization procedures (CABG *n* = 2, PCI *n* = 1), of whom 2 were pre-operative patients (CABG) and 1 patient had heart failure (PCI).

### 3.5. Mortality

All-cause mortality during the 6-month follow-up was 5.7% (*n* = 5) in the patients that underwent direct CAG and 2.4% (*n* = 8) in the patients that underwent initial CTCA (*p* = 0.12). In the patients that underwent initial CAG, all-cause mortality was highest in the patients with obstructive CAD (12.5%) and lower for patients with non-obstructive CAD (1.4%). In patients that underwent CTCA, all-cause mortality was highest in those with obstructive CAD on CTCA (5.3%) and lower for the patients with non-obstructive CAD (3.5%) (*p* = 0.03). All-cause mortality was 0% in the patients in which CTCA indicated no signs of CAD.

## 4. Discussion

In this study, we evaluated the implementation of initial CTCA in the diagnostic work-up for non-coronary cardiac surgery, cardiomyopathy, heart failure and ventricular arrhythmias. The main findings of our study were that, firstly, a diagnostic strategy of initial CTCA was performed in 327/415 patients (78.8%). Secondly, in patients undergoing initial CTCA, additional CAG was clinically indicated in only 16.8%, and coronary lesions of > 50% DS were confirmed by CAG in only 21 out of 55 patients (38.2%, 6.3% of total). Thirdly, in no patients did CTCA underestimate the degree of stenosis compared with angiographic assessment. Fourthly, in patients referred directly to CAG by the ordering physician’s preference, 17.2% had coronary lesions of > 50% DS. Finally, coronary revascularization was only performed in 3.0% of patients that underwent initial CTCA and in 3.4% of patients that underwent direct CAG.

In this study, we showed that implementation of initial CTCA can reduce the need for invasive CAG by 83% in the diagnostic work-up for non-coronary cardiac surgery, cardiomyopathy, heart failure and ventricular arrhythmias. Previous studies with a similar design reported a reduction in CAG, ranging between 87.9% for patients with chest pain and 76.4% in patients referred for transcatheter aortic valve procedures [7,8,9]. Accordingly, in a cohort of 398,978 patients who underwent elective CAG, obstructive CAD was diagnosed in only one third of patients [10]. These results suggest that the majority of CAGs can be avoided in a wide range of indications and that obstructive CAD requiring revascularization is not diagnosed frequently. In our study in patients without typical angina symptoms, additional CAG was indicated in only 55 patients (16.8%). In patients who underwent both initial CTCA and additional CAG, the degree of stenosis was equal, or CTCA overestimated the stenosis. There were no cases in which CTCA underestimated the degree of stenosis. This observation confirms previous research showing that CTCA tends to overestimate the degree of stenosis [11]. This high diagnostic sensitivity and negative predictive value of CTCA make it highly suitable as a “gatekeeper” for invasive CAG, particularly in patients with a low to moderate pre-test likelihood of obstructive CAD. Furthermore, the number of invasive CAG procedures could potentially be reduced further with the addition of CT-FFR or non-invasive functional imaging as a secondary diagnostic step, preserving invasive CAG for the patients with substantial myocardial ischemia. Additionally, the ISCHEMIA trial showed that initial optimal medical therapy is appropriate, even in patients with obstructive CAD and substantial myocardial ischemia, raising the question if diagnostic evaluation should be performed at all [12]. However, for the patient population presented in our study, future research will have to show whether this also applies to patients with reduced LV function, valvular disease, ventricular arrhythmias or cardiomyopathy, as these were not included in that trial.

Despite implementing initial CTCA for the aforementioned indications, not all patients were referred to initial CTCA. A total of 88 patients were referred to direct CAG as deemed by the treating cardiologist. These patients were older and had more cardiovascular comorbidities than the patients that underwent initial CTCA. Remarkably, the percentage of patients with lesions of >50% DS in the direct CAG group was also low (17.2%). Therefore, performing initial CTCA in these patients would also have led to a reduction in the number of CAGs performed.

Challenges for successful implementation of CTCA include general factors such as organizational culture, financial resources, the availability of technically advanced CT scanners and education and training of staff [13,14]. During this project, we learned that it is necessary to involve all relevant disciplines: general cardiology, cardiac imaging and interventional cardiology, radiology and cardiothoracic surgery. Secondly, we learned that some physicians preferred direct CAG over initial CTCA because these physicians were more comfortable with the known modality (CAG). Therefore, we invested in a dedicated physician to coordinate the implementation of CTCA as an initial test. This physician encouraged the other physicians to refer patients for initial CTCA and evaluated all CAG orders. To familiarize the physicians with CTCA, we held recurrent presentations to inform physicians about the current image quality, CTCA–CAG correlations and general capabilities of CTCA and shared the provisional results.

The implementation of initial CTCA for the proposed patient groups will increase the demand for CTCA services substantially. This may pose challenges for the availability of CTCA services, and hospitals may have to increase CTCA capacity, which includes investing in additional CTCA-capable CT scanners or cardiovascular imaging experts. This becomes an important issue in small community hospitals where limited resources have to be used efficiently. Furthermore, these hospitals may not be able to set up a profitable CTCA program next to the existing CAG program, which also needs sufficient resources. National or regional governance or initiatives may be required to ensure a smooth transition towards CTCA healthcare, managed by all involved medical specialties and stakeholders.

### Limitations

First of all, our results should be perceived as the results of a non-randomized, single-center cohort study in an academic setting performed on a state-of-the-art CT scanner. In this center, there is extensive knowledge of CTCA acquisition, contrast delivery protocols and image reading. Therefore, the diagnostic yield found in this trial may be higher than in clinical practice with a lower-end CT scanner or different image acquisition protocols. Nevertheless, we believe that a high percentage of additional CAGs can be avoided with the implementation of initial CTCA. Secondly, we informed all cardiologists at the start of this project about its implementation and actively encouraged them to refer patients to initial CTCA. However, the choice to refer patients to initial CTCA or direct CAG was at the discretion of the attending physician. The patients that were referred to direct CAG were older and had more comorbidities than the patients that were referred to initial CTCA. These factors could, at least in theory, have affected the number of non-diagnostic CTCA examinations and the prevalence of obstructive CAD.

## 5. Conclusions

The use of CTCA as a gatekeeper CAG in the diagnostic work-up of non-coronary cardiac surgery, cardiomyopathy, heart failure and ventricular arrhythmias is feasible, and only 17% of the patients in this study required additional CAG after initial CTCA. Therefore, CTCA should be considered as the initial imaging modality to rule out CAD in these patients.

## Figures and Tables

**Table 1 jcm-10-02374-t001:** Baseline characteristics of patients with initial CTCA, according to indications.

	Total	Preoperative	Cardiomyopathy	*p*-Value	Heart Failure	*p*-Value	Ventricular Arrhythmia	*p*-Value
Number of patients	327	163	57		31		76	
Patient characteristics								
- Age, years (SD)	57.9 (14.0)	59.3 (14.4)	55.6 (13.8)	0.09	61.6 (15.0)	0.36	55.1 (12.3)	0.03
- Male gender, *n* (%)	220 (67.3)	108 (66.3)	40 (70.2)	0.71	18 (58.1)	0.50	54 (71.1)	0.56
- BMI, kg/m^2^ (SD)	26.2 (4.7)	26.1 (4.9)	26.5 (4.0)	0.59	25.5 (4.2)	0.53	26.4 (5.2)	0.66
Cardiovascular comorbidities								
- Atrial fibrillation, *n* (%)	77/289 (26.6)	46 (30.9)	13 (27.1)	0.75	9 (33.3)	0.98	9 (13.6)	0.01
- Impaired renal function, *n* (%)	43/319 (13.5)	23 (14.6)	10 (18.2)	0.67	9 (29.0)	0.09	1 (1.3)	0.004
- Diabetes mellitus, *n* (%):	35/285 (12.3)	14 (10.0)	8 (15.4)	0.43	7 (25.0)	0.06	6 (9.4)	1.00
- Dyslipidemia, *n* (%)	98/288 (34.0)	47 (34.3)	17 (35.0)	1.00	16 (57.7)	0.05	18 (23.4)	0.19
- Hypertension, *n* (%)	122/306 (39.9)	66 (42.9)	18 (34.0)	0.33	17 (54.8)	0.31	21 (30.9)	0.13
- Current smoker, *n* (%)	58/284 (20.4)	22 (15.1)	15 (31.9)	0.02	10 (37.0)	0.02	11 (16.9)	0.89
- Family history of CAD, *n* (%)	106/250 (42.4)	56 (46.7)	21 (45.7)	1.00	6 (26.1)	0.11	23 (37.7)	0.32

CTCA: computed tomography coronary angiography, SD: standard deviation, BMI: body mass index. Clinical characteristics of the indications with a smaller number of patients (cardiomyopathy, heart failure, ventricular arrhythmia) were compared with the patients awaiting non-coronary cardiac surgery.

**Table 2 jcm-10-02374-t002:** Indications and results of the patients that underwent initial CTCA.

Patient Group	Initial CTCA	CAG after Initial CTCA	Direct CAG
Number of patients, *n* (%)	327	55 (16.8)	88
Indication, *n* (%)			
- Preoperative	163 (49.8)	35 (63.6)	32 (36.4)
- Cardiomyopathy	57 (17.4)	6 (10.9)	26 (29.5)
- Heart failure	31 (9.5)	5 (9.1)	24 (27.3)
- Ventricular arrhythmias	76 (23.2)	9 (16.4)	6 (6.8)
Diameter stenosis, *n* (%)			
- <50%	260 (79.5)	34 (61.8)	72 (81.8)
- 50–70%	30 (9.2)	10 (18.2)	10 (11.5)
- >70%	23 (7.0)	11 (20.0)	6 (6.8)
- Non-diagnostic	14 (4.3)	0 (0.0)	0 (0.0)

Indications and results of patients that underwent initial CTCA, those with an additional CAG after CTCA and patients directly referred to CAG. CTCA: computed tomography coronary angiography, CAG: coronary angiography.

**Table 3 jcm-10-02374-t003:** Baseline characteristics of patients that were referred to first-line CTCA or direct CAG.

Patient Group	Initial CTCA	Direct CAG	*p*-Value
Number of patients	327	88	
Patient characteristics			
- Age, years ± SD	57.9 ± 14.0	66.0 ± 10.5	<0.001
- Male gender, *n* (%)	220 (68.3)	52 (59.1)	0.19
- BMI, kg/m^2^ ± SD	26.2 ± 4.7	27.5 ± 4.9	0.03
- Mean pre-test risk	11.1 ± 8.6	13.7 ± 8.2	<0.001
Cardiovascular comorbidities			
- Atrial fibrillation, *n* (%)	77/289 (26.6)	29/85 (34.1)	0.23
- Impaired renal function, *n* (%)	43/319 (13.5)	23/87 (26.4)	0.01
- Diabetes mellitus, *n* (%):	35/285 (12.3)	15/84 (17.9)	0.26
- Dyslipidemia, *n* (%)	98/288 (34.0)	25/73 (34.3)	1.00
- Hypertension, *n* (%)	122/306 (39.9)	39/84 (46.4)	0.34
- Current smoker *n* (%)	58/284 (20.4)	16/82 (19.5)	1.00
- Family history of CAD, *n* (%)	106/250 (42.4)	32/69 (46.4)	0.65

Baseline characteristics of patients that were referred to first-line CTCA or direct CAG. CTCA: coronary computed tomography angiography, CAG: coronary angiography, SD: standard deviation, BMI: body mass index.

## Data Availability

Not applicable.

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
