# Peer review of "Implementation of CT Coronary Angiography as an Alternative to Invasive Coronary Angiography in the Diagnostic Work-Up of Non-Coronary Cardiac Surgery, Cardiomyopathy, Heart Failure and Ventricular Arrhythmias"

_jcm, 2021, doi:10.3390/jcm10112374_

Round 1

Reviewer 1 Report

This is a simple observational analysis of CVD patients undergone CTCA to identify if ICA is necessary. The work is well performed and the manuscript reads well. Some minor revisions are proposed by the reviewer: 

1) the novelty of the work is not clear. 

2) some state of the art work related to this study is missing. For example the authors must discuss their findings with the results from ISCHEMIA study. 

3) It is also useful to discuss about the additional effect that hemodynamics may have to identify patients who really need ICA. For example the non-invascive FFR or the shear stress calculation. There are plenty of such works from heart flow company, from Clemente et al. Sakellarios et al. and other. Also the added value of perfusion CT or PET imaging could be also discussed. 

4) Finally, I would like to see whether the CTCA imaging findings (degree of stenosis, lumen area, plaques, lesions) can be used to predict the need of ICA. 

Author Response

We thank the reviewer for reviewing our manuscript entitles “Implementation of CT-coronary angiography as an alternative to invasive coronary angiography in the diagnostic work-up of non-coronary cardiac surgery, cardiomyopathy, heart failure, and ventricular arrhythmias”. We respond to the comments raised.

1) the novelty of the work is not clear. 

The current ESC guidelines focus on patients presenting to the cardiologist with chest pain and suspected CAD. In these guidelines, the recommendation is specifically focused on six frequently encountered clinical scenarios: (i) patients with suspected CAD and ‘stable’ anginal symptoms, and/or dyspnea, (ii) patients with new onset of heart failure or left ventricular dysfunction and suspected CAD, (iii) asymptomatic and symptomatic patients with stabilized symptoms <1 year after an ACS, or patients with recent revascularization, (iv) asymptomatic and symptomatic patients >1 year after initial diagnosis or revascularization, (v) patients with angina and suspected vasospastic or microvascular disease, and (vi) asymptomatic subjects in whom CAD is detected at screening.

In the clinical scenarios for patients in the work-up of non-coronary cardiac surgery, cardiomyopathy, and heart failure, CTCA is mentioned in the guideline as an alternative for invasive CAG, only in patients with a low risk for CAD. However, the paradigm in clinical practice is that these patients have high risk for CAD and is an underlying cause of the condition in patients with cardiomyopathy and heart failure or a frequent comorbidity in the patients valvular disease (with non-coronary cardiac surgery). Therefore, CTCA is not considered routinely. Our hypothesis was that implementation of CTCA as gatekeeper for CAG in these patient groups was feasible and that only a minority of patients would require additional CAG.

2) some state of the art work related to this study is missing. For example the authors must discuss their findings with the results from ISCHEMIA study. 

We agree with the reviewer that the ISCHEMIA trial should be regarded as a landmark trial for patients in the work-up with angina pectoris and obstructive CAD. In this trial, all patients with obstructive CAD on CTCA underwent functional testing and, in case of substantial ischemia, the patients underwent initial optimal medical therapy versus direct revascularization. In this trial, initial optimal medical therapy was non-inferior, compared to direct revascularization. In contrast to the ISCHEMIA trial, the patients in our submitted trial are in the work-up for non-coronary cardiac surgery, cardiomyopathy, heart failure, and ventricular arrhythmias. Therefore, the reason to consider diagnostic evaluation is different, compared to the patients presenting with chest pain. In the presented results, CTCA can be considered as a substitute for invasive CAG, whereas CTCA is the initial test in the ISCHEMIA trial, as compared to i.e. exercise ECG or SPECT. Furthermore, the ISCHEMIA specifically did not include patients with reduced LV function, valvular disease or cardiomyopathy.

However, the ISCHEMIA trial showed that a more restrictive approach towards revascularization can be considered. Therefore, we adjusted the discussion and included this lesson learned from the ISCHEMIA trial.

The added sentences read:

Furthermore, the number of invasive CAG procedures could potentially be reduced further with the addition of CT-FFR or non-invasive functional imaging as secondary diagnostic step, preserving invasive CAG for the patients with substantial myocardial ischemia. Ad-additionally, the ISCHEMIA trial showed that initial optimal medical therapy is appropriate, even in patients with obstructive CAD and substantial myocardial ischemia, raising the question if diagnostic evaluation should be performed at all [12]. However, for the patient population presented in our study, future research will have to show whether this also applies to patients with reduced LV function, valvular disease,  ventricular arrhythmias or cardiomyopathy, as these were not included in that trial.

3) It is also useful to discuss about the additional effect that hemodynamics may have to identify patients who really need ICA. For example the non-invasive FFR or the shear stress calculation. There are plenty of such works from heart flow company, from Clemente et al. Sakellarios et al. and other. Also the added value of perfusion CT or PET imaging could be also discussed. 

We have included CT-FFR or non-invasive functional testing in the discussion.

Furthermore, the number of invasive CAG procedures could potentially be reduced further with the addition of CT-FFR or non-invasive functional imaging as secondary diagnostic step, preserving invasive CAG for the patients with substantial myocardial ischemia.

4) Finally, I would like to see whether the CTCA imaging findings (degree of stenosis, lumen area, plaques, lesions) can be used to predict the need of ICA. 

In this clinical study we evaluated all coronary segments with a diameter >1.5 mm for the presence of coronary stenosis and graded according to the standardized Coronary Artery Disease - Reporting and Data System (CAD-RADS) method. The need for ICA was dependent on the findings on CTCA in which a ICA was performed in case of CAD-RADS 4 or higher. In some cases in which there was CAD-RADS 3 and arguable involvement of the proximal LAD, a ICA was performed for additional visual or hemodynamic (FFR) evaluation. The CAD-RADS score includes degree of stenosis or lumen area. 

Reviewer 2 Report

Excellent original study

Author Response

We thank the reviewer for the appreciation of our work.

Reviewer 3 Report

In the present study the authors analysed the need for additional invasive CAG after initial CTCA in patients awaiting non-coronary cardiac surgery and in patients with cardiomyopathy, heart failure or ventricular arrhythmias. Patients were categorized as those referred to initial CTCA (conform protocol), and to direct CAG (non-conform protocol). Out of a total of 415 patients, 78.8% were referred to initial CTCA, of whom 260 patients had no obstructive lesions. A total of 55 patients (16.8%) underwent additional CAG after initial CTCA, which showed coronary lesions of >50%  in 21 patients (6.3% of 327). Eighty-eight patients were directly referred to CAG (non-conform protocol). These patients were older and had more cardiovascular risk factors, and coronary lesions >50% DS were detected in 16 patients (17.2%). Revascularization procedures were low: initial CTCA (3.0%), direct CAG (3.4%).  The authors concluded that the use of CTCA in this setting is feasible and only 17% of these patients required additional CAG after initial CTCA. Therefore, CTCA should be considered as the initial imaging modality to rule out CAD in these patients.

Major comment

This an interesting study focused substantially on the implementation of CTCA scan as alternative method to invasive CAG in setting of non-coronary cardiac surgery, cardiomyopathy, heart failure or ventricular arrhythmias.  Some questions should be clarified to better understand results of this study.

First, the adoption of conform and non-conform protocol is at discretion of attending physician. Really, the baseline cardiovascular risk profile of both 2 groups patients should be showed, and, according to ESC guidelines, patients with low-intermediate risk profile are ideal candidate to coronary CTCA scan. At contrary, patients in non-conform protocol, being holder and with more CV risk factors, as a rule should be evaluated with invasive CAG. In addition, the functional significance of obstructive CAD was not assessed in these asymptomatic patients, thus the revascularization may be coronary “stenotic-oriented”. Second, in clinical practice the presence of non-obstructive CAD may be associated with a poor outcome in comparison to the absence of CAD, and generally a statin agent is recommended to avoid events in long term. In setting of non-coronary cardiac surgery, no data were reported. It is conceivable a protective role of statin agent for non-obstructive coronary plaques also in this setting. Third, from methodological point of view, the recruitment from 2017 to 2020, may imply the use of different protocol and different CTCA scan readers, each of one potential confounders of results obtained. Finally, the authors did not provided data about the reliability of CTCA scan.

With all this in mind, to improve the value of the study the authors should:

  • Classify patients of both groups according to the low-intermediate-high risk profile of new ESC  recommendation (2019),  
  • Classify the coronary plaques other than the degree of stenosis, as calcified, not calcified or mixed
  • better explains the inclusion criteria of atrial fibrillation patients and the method used to avoids motion artefacts
  • Explain, why after initial CCTA, no patients underwent functional test before CAG
  • Explain  the criteria adopted for revascularization (detection of myocardial ischemia, FFR, niFFR-CT, CT perfusion, etc)
  • Provide medical treatment recommended during the intervention for patients showing not-obstructive CAD
  • Show differences, if any, on the outcome between non-obstructive CAD vs. absence of CAD
  • Show difference, if any, on the outcome between revascularized vs. no-revascularized patients.
  • Provide the reliability of CT scan
  • Try to discuss the value of functional test in this setting, especially for patients presenting contraindications for CTCA

Author Response

Major comment

This an interesting study focused substantially on the implementation of CTCA scan as alternative method to invasive CAG in setting of non-coronary cardiac surgery, cardiomyopathy, heart failure or ventricular arrhythmias.  Some questions should be clarified to better understand results of this study.

We thank the reviewer for the appreciation of our work and will try to include all revisions and to clarify all questions raised.

First, the adoption of conform and non-conform protocol is at discretion of attending physician. Really, the baseline cardiovascular risk profile of both 2 groups patients should be showed, and, according to ESC guidelines, patients with low-intermediate risk profile are ideal candidate to coronary CTCA scan. At contrary, patients in non-conform protocol, being holder and with more CV risk factors, as a rule should be evaluated with invasive CAG. In addition, the functional significance of obstructive CAD was not assessed in these asymptomatic patients, thus the revascularization may be coronary “stenotic-oriented”. Second, in clinical practice the presence of non-obstructive CAD may be associated with a poor outcome in comparison to the absence of CAD, and generally a statin agent is recommended to avoid events in long term. In setting of non-coronary cardiac surgery, no data were reported. It is conceivable a protective role of statin agent for non-obstructive coronary plaques also in this setting. Third, from methodological point of view, the recruitment from 2017 to 2020, may imply the use of different protocol and different CTCA scan readers, each of one potential confounders of results obtained. Finally, the authors did not provided data about the reliability of CTCA scan.

With all this in mind, to improve the value of the study the authors should:

  • Classify patients of both groups according to the low-intermediate-high risk profile of new ESC recommendation (2019) 

We agree with the reviewer that this is a valuable addition. As the patients did not have angina complaints, the maximum risk score in the ESC guidelines table is 24% and all patients are low to intermediate risk. The mean risk profiles were added and compared in table 3

  • Classify the coronary plaques other than the degree of stenosis, as calcified, not calcified or mixed

In this clinical study we evaluated all coronary segments with a diameter >1.5 mm for the presence of coronary stenosis and graded according to the standardized Coronary Artery Disease - Reporting and Data System (CAD-RADS) method. The need for ICA was dependent on the findings on CTCA in which a ICA was performed in case of CAD-RADS 4 or higher. In some cases in which there was CAD-RADS 3 and arguable involvement of the proximal LAD, a ICA was performed for additional visual or hemodynamic (FFR) evaluation. The CAD-RADS score combines the degree of stenosis or lumen area and plaque specifications together in a score, used for clinical decision making.

  • better explains the inclusion criteria of atrial fibrillation patients and the method used to avoids motion artefacts

Patients with atrial fibrillation were included in the study. Before CTCA, oral beta blockers were administered if the heart rate was >65 per min. The patients underwent CTCA with an alternative CTCA acquisition protocol in case of arrhythmias. A high pitch spiral scanning was performed in diastole in patients with regular heart rate <70/min. For patients with irregular heart rates or heart rates >70/min, a prospective sequential scan was performed in diastole and for heart rates >80/min in systole.

We have adjusted the methods section accordingly:

This study included patients that would typically undergo invasive CAG before the start of the project. If CTCA indicated obstructive CAD, a subsequent CAG was performed. Atrial fibrillation was no exclusion criterion. However, the CTCA acquisition protocol was adjusted accordingly.

  • Explain, why after initial CCTA, no patients underwent functional test before CAG

We agree with the reviewer that subsequent functional testing would further reduce the number of invasive CAG. However, especially in the patients in the work-up for non-coronary cardiac surgery, anatomical diagnostic evaluation is performed to visualize anatomical high risk lesions, that can potentially be revascularized with coronary artery bypass grafting during surgery. We have added sentences in the methods that clarify that CTCA, in this project, was used as a substitution of invasive CAG. In the light of the ISCHEMIA trial, the paradigm had shifted towards a diagnostic work-up of CTCA and functional testing before considering revascularization in patients with CAD. However, the ISCHEMIA trial did not include patients with LV dysfunction, valvular disease, cardiomyopathy or arrhythmias.

We have added the sentence below to the methods:

This study included patients that would typically undergo invasive CAG before the start of the project. If CTCA indicated obstructive CAD, a subsequent CAG was performed.

  • Explain the criteria adopted for revascularization (detection of myocardial ischemia, FFR, niFFR-CT, CT perfusion, etc)

We have added the criteria in the methods section. This now reads:

Revascularization was at the discretion of the interventional cardiologist, based on invasive CAG with fractional flow reserve (FFR).

  • Provide medical treatment recommended during the intervention for patients showing not-obstructive CAD

Although medical treatment was not prospectively collected for the purpose of the current study, in our center we generally recommend prescribing a statin and aspirin in case of non-obstructive CAD on CTCA.

  • Show differences, if any, on the outcome between non-obstructive CAD vs. absence of CAD

We agree with the reviewer that this is valuable information and added in to the results

The section now reads:

All-cause mortality during the 6-month follow-up was 5.7% (n=5) in the patients that underwent direct CAG and 2.4% (n=8) in the patients that underwent initial CTCA (p=0.12). In the patients that underwent initial CAG, all-cause mortality was highest in the patients with obstructive CAD (12.5%) and lower for patients with non-obstructive CAD (1.4%). In patients that underwent CTCA, all-cause mortality was highest in those with obstructive CAD on CTCA (5.3%), and lower for the patients with non-obstructive CAD (3.5%) (p=0.03). All-cause mortality was 0% in the patients in which CTCA indicated no signs of CAD.

  • Show difference, if any, on the outcome between revascularized vs. no-revascularized patients

We agree with the reviewer that this is an interesting topic. However, our sample size and the low occurrence of mortality in this trial does not enable a thorough subgroup analysis. Furthermore, the patients that were revascularized were revascularized for different reasons, making the discussion regarding retrospective analysis on this topic difficult. However we would like to share the information regarding mortality in these subgroups with the reviewer.

Taking both the patients that underwent initial CAG and initial CTCA together. The mortality in the patients with obstructive CAD that underwent revascularization was 7.7% (n=1 out of 13) vs. 6.7% (n=4 out of 60) in the patients with obstructive CAD that did not underwent revascularization. The other deaths occurred in the patients with non-obstructive CAD 3.8% (8 out of 213)

  • Provide the reliability of CT scan

We agree with the reviewer that this would be valuable information. However, as this was a clinical project, we did not include additional CT-readers. Therefore, we cannot analyze the inter observer agreement of out CT-scans.

  • Try to discuss the value of functional test in this setting, especially for patients presenting contraindications for CTCA

We have included a discussion on this topic in the discussion. This section now reads:

Furthermore, the number of invasive CAG procedures could potentially be reduced further with the addition of CT-FFR or non-invasive functional imaging as secondary diagnostic step, preserving invasive CAG for the patients with substantial myocardial ischemia. In patient with contraindication for CTCA, a functional test can be considered as initial test. Additionally, the ISCHEMIA trial showed that initial optimal medical therapy is appropriate, even in patients with obstructive CAD and substantial myocardial ischemia, raising the question if diagnostic evaluation should be performed at all [12]. However, for the patient population presented in our study, future research will have to show whether this also applies to patients with reduced LV function, ventricular arrhythmias or cardiomyopathy, as these were not included in that trial.

Reviewer 4 Report

New 2019 ESC Guidelines need to be worked into the paper.

Coronary CT is already an accepted tool for exclusion of CAD.

Rate of interventions is the same in primary CAG and Cardiac CT group - it shouldn't be - please explain

Prospective CT in arrythmic patients should not be done, pleas clarify which arrythmia?

Why is there a so high of non diagnostic cardiac CT (>10%), please clarify

Author Response

Reviewer 4

New 2019 ESC Guidelines need to be worked into the paper.

Coronary CT is already an accepted tool for exclusion of CAD.

The current ESC guidelines focus on patients presenting at the cardiologist with chest pain and suspected CAD. In these guidelines, the recommendation specifically focus on six frequently encountered clinical scenarios: (i) patients with suspected CAD and ‘stable’ anginal symptoms, and/or dyspnea, (ii) patients with new onset of heart failure or left ventricular dysfunction and suspected CAD, (iii) asymptomatic and symptomatic patients with stabilized symptoms <1 year after an ACS, or patients with recent revascularization, (iv) asymptomatic and symptomatic patients >1 year after initial diagnosis or revascularization, (v) patients with angina and suspected vasospastic or microvascular disease, and (vi) asymptomatic subjects in whom CAD is detected at screening.

In the clinical scenarios for patients in the work-up of non-coronary cardiac surgery, cardiomyopathy, and heart failure, CTCA is mentioned as an alternative for invasive CAG in patients, only with a low risk for CAD. However, the paradigm in clinical practice is that these patients have high risk for CAD and is an underlying cause of the condition in patients with cardiomyopathy and heart failure or a frequent comorbidity in the patients valvular disease (with non-coronary cardiac surgery). Therefore, CTCA is not considered routinely. Our hypothesis was that implementation of CTCA as gatekeeper for CAG was feasible in all these patient groups and that only a minority of patients would require additional CAG.

Rate of interventions is the same in primary CAG and Cardiac CT group - it shouldn't be - please explain

In both groups that underwent initial CTCA and initial CAG, the incidence of patients without obstructive CAD is high. In the remaining patients with obstructive CAD, the percentage of patients that underwent revascularization was similar in both groups. We agree with the reviewer that this is an interesting topic. However, both CTCA and CAG are anatomical diagnostic tests, visualizing the extend or CAD. The method of diagnosing anatomical obstructive CAD does not influence the rate of revascularization. We agree with the reviewer that it is expected that certain methods will influence the number of revascularizations. Like the ISCHEMIA trial, work-up with non-invasive functional imaging is likely to reduce the need for revascularization.

Prospective CT in arrhythmic patients should not be done, pleas clarify which arrythmia?

Patients with atrial fibrillation were included in the study. Before CTCA, oral beta blockers were administered if the heart rate was >65 per min. The patients underwent CTCA with an alternative CTCA acquisition protocol in case of arrhythmias. A high pitch spiral scanning was performed in diastole in patients with regular heart rate <70/min. For patients with irregular heart rates or heart rates >70/min, a prospective sequential scan was performed in diastole and for heart rates >80/min in systole.

Why is there a so high of non-diagnostic cardiac CT (>10%), please clarify

The rate of non-diagnostic CTCA in the presented data was 4,3% (14 out of 327)

Round 2

Reviewer 3 Report

Some questions remain matter of debate. I suggest adding in the
limitation and conlusion sections that although promising, caution is
needed before extending these findings into clinical practice,
especially in community hospitals,
which often lack a CT scan or
adequate medical experience for reading.

Author Response

We thank the reviewer for the extensive review and the comments raised.  

We agree with the reviewer that our high diagnostic yield of CTCA was the result of an excellent CT-scanner, with the addition of experienced personnel and adequate image acquisition protocols. However, we believe that sharing these results may facilitate implementation of CTCA in these patient groups and helps to modernize clinical practice, also in community hospitals. Nevertheless, we agree with the reviewer, that although promising, caution is needed before extending these findings into clinical practices without modern CT-scanner, extensive knowledge of CTCA acquisition and contrast delivery protocols. We have changed the limitation section appropriately and now reads: 

First of all, our results should be perceived as results of a non-randomized, single-centre cohort study in an academic setting performed on a state-of-the-art CT scanner. In this centre, there is extensive knowledge of CTCA acquisition, contrast delivery protocols and image reading. Therefore, the diagnostic yield found in this trial may be higher than in clinical practice with a lower-end CT-scanner or different image acquisition protocols. Nevertheless, we believe that a high percentage of additional CAG’s can be avoided with the implementation of initial CTCA. Secondly, we informed all cardiologists at the start of this project about its implementation and actively encouraged them to refer patients to initial CTCA. However, the choice to refer patients to initial CTCA or direct CAG was at the discretion of the attending physician. The patients that were referred to direct CAG were older and had more comorbidities than the patients that were referred to initial CTCA. These factors could, at least in theory, have affected the number of non-diagnostic CTCA examinations and the prevalence of obstructive CAD.